# Simple Black-Box Adversarial Perturbations for Deep Networks

**Nina Narodytska**     **Shiva Kasiviswanathan**
Samsung Research America
Mountain View, CA 94043, USA
{n.narodytska,kasivisw}@gmail.com

## Abstract

Deep neural networks are powerful and popular learning models that achieve state-of-the-art pattern recognition performance on many computer vision, speech, and language processing tasks. However, these networks have also been shown susceptible to carefully crafted adversarial perturbations which force misclassification of the inputs. Adversarial examples enable adversaries to subvert the expected system behavior leading to undesired consequences and could pose a security risk when these systems are deployed in the real world.

In this work, we focus on deep convolutional neural networks and demonstrate that adversaries can easily craft adversarial examples even without any internal knowledge of the target network. Our attacks treat the network as an oracle (black-box) and only assume that the output of the network can be observed on the probed inputs. Our first attack is based on a simple idea of adding perturbation to a randomly selected single pixel or a small set of them. We then improve the effectiveness of this attack by carefully constructing a small set of pixels to perturb by using the idea of greedy local-search. Our proposed attacks also naturally extend to a stronger notion of misclassification. Our extensive experimental results illustrate that even these elementary attacks can reveal a deep neural network's vulnerabilities. The simplicity and effectiveness of our proposed schemes mean that they could serve as a litmus test for designing robust networks.

## 1 Introduction

Convolutional neural networks (CNNs) are among the most popular techniques employed for computer vision tasks, including but not limited to image recognition, localization, video tracking, and image and video segmentation (Goodfellow et al., 2016). Though these deep networks have exhibited good performances for these tasks, they have recently been shown to be particularly susceptible to adversarial perturbations to the input images (Szegedy et al., 2014; Goodfellow et al., 2015; Moosavi-Dezfooli et al., 2016; Papernot et al., 2016c;b; Kurakin et al., 2016; Grosse et al., 2016; Zagoruyko, 2016b). Vulnerability of these networks to adversarial attacks can lead to undesirable consequences in many practical applications using them. For example, adversarial attacks can be used to subvert fraud detection, malware detection, or mislead autonomous navigation systems (Papernot et al., 2016c; Grosse et al., 2016). Further strengthening these results is a recent observation by Kurakin et al. (2016) who showed that a significant fraction of adversarial images crafted using the original network are misclassified even when fed to the classifier through a physical world system (such as a camera).

In this paper, we investigate the problem of robustness of state-of-the-art convolutional neural networks (CNNs) to simple black-box adversarial attacks. The rough goal of adversarial attacks is as follows: Given an image $I$ that is correctly classified by a machine learning system (say, a CNN), is it possible to construct a transformation of $I$ (say, by adding a small perturbation to some or all the pixels) that now leads to misclassification by the system. Since large perturbations can trivially lead to misclassification, the attacks seek to limit the amount of perturbation applied under some chosen metric. More often than not, in these attacks, the modification done to the image is so subtle that the changes are imperceptible to a human eye. Our proposed attacks also share this property, in addition to being practical and simplistic, thus highlighting a worrying aspect about lack of robustness prevalent in these modern vision techniques.

There are two main research directions in the literature on adversarial attacks based on different assumptions about the adversarial knowledge of the target network. The first line of work assumes that the adversary has detailed knowledge of the network architecture and the parameters resulting from training (or access to the labeled training set) (Szegedy et al., 2014; Goodfellow et al., 2015; Moosavi-Dezfooli et al., 2016; Papernot et al., 2016c). Using this information, an adversary constructs a perturbation for a given image. The most effective methods are gradient-based: a small perturbation is constructed based on the gradients of the loss function w.r.t. the input image and a target label. Often, adding this small perturbation to the original image leads to a misclassification. In the second line of work an adversary has restricted knowledge about the network from being able to only observe the network's output on some probed inputs (Papernot et al., 2016b). Our work falls into this category. While this *black-box* model is a much more realistic and applicable threat model, it is also more challenging because it considers weak adversaries without knowledge of the network architecture, parameters, or training data. Interestingly, our results suggest that this level of access and a small number of queries provide sufficient information to construct an adversarial image.

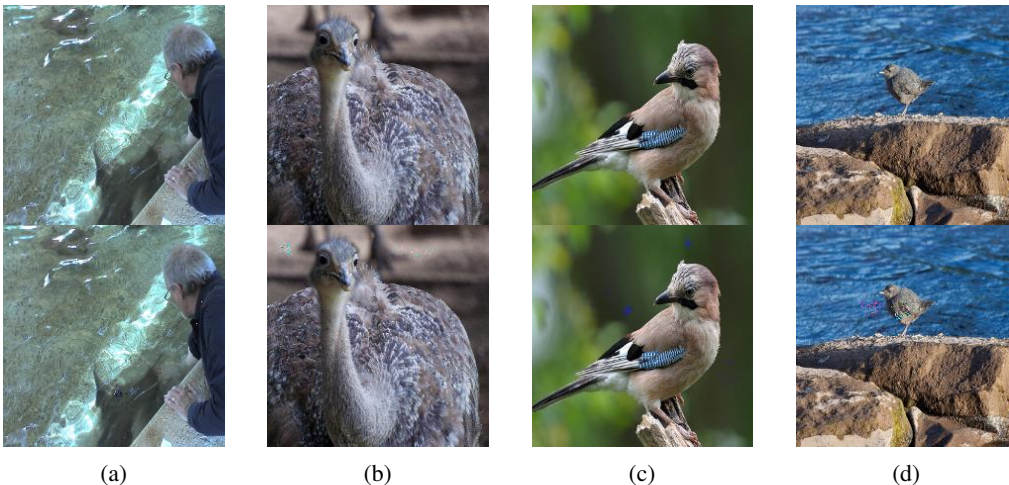

|              |              |              |              |
| :----------: | :----------: | :----------: | :----------: |
| (a)          | (b)          | (c)          | (d)          |

Table 1: The top row shows the original images and the bottom row shows the perturbed images. The misclassification is as follows: (a) a stingray misclassified as a sea lion, (b) an ostrich misclassified as a goose, (c) a jay misclassified as a junco, and (d) a water ouzel misclassified as a redshank.

As we operate in a black-box setting, we use a gradient-free approach to adversarial image generation. Papernot et al. (2016b) were the first to discuss a black-box attack against deep learning systems. Their attack crucially relies on the observation that there is a *transferability* (*generalization*) property in adversarial examples, i.e., adversarial examples form one model transfers to another. Our proposed attacks on the other hand is much more simple and direct, does not require this transferability property, and hence is more effective in constructing adversarial images, in addition to having some other computational advantages. We demonstrate that our method is capable of constructing adversarial images for several network architectures trained on different datasets. In particular in this paper, we consider the CIFAR10, MNIST, SVHN, STL10, and ImageNet1000 datasets, and two popular network architectures, Network-in-Network (Lin et al., 2014) and VGG (Simonyan & Zisserman, 2014). In Table 1, we show four images from the ImageNet1000 dataset. The original images are in the upper row. The bottom row shows the corresponding perturbed images produced by our algorithm which are misclassified by a VGG CNN-S network (Chatfield et al., 2014a).

**Our Contributions.**    In this work, we present simple and effective black-box adversarial attacks on deep convolutional neural networks. We make the following main contributions in this paper.

**(1)** The first question we investigate is the influence of perturbing a *single* pixel on the prediction. To do so, we devise a simple scheme, based on randomly selecting a single pixel and applying a strong perturbation to it. Somewhat surprisingly, we noticed that a few trails of this random experiment is already quite enough in generating adversarial images for low resolution image sets. In fact, in many cases, for misclassification, the amount of perturbation needed to be applied to the

selected pixel is also quite small. For high-resolution images, a similar phenomena holds, except our scheme now picks a random set of around 50 pixels. These simple experiments show the ease of generating adversarial images for modern deep CNNs without knowledge of either the network architecture or its parameters. There is however one shortcoming in these approaches in that the perturbed image might have pixel values that are outside some expected range.

**(2)** We overcome this above shortcoming by showing that lower perturbation suffices if we carefully select the pixels for perturbation. The approach is based the idea of *greedy local search*, an iterative search procedure, where in each round a local neighborhood is used to refine the current image and in process minimizing the probability of the network assigning high confidence scores to the true class label. Again while the algorithm is quite simple, it is rather effective in generating adversarial images with quite small perturbations. We also show an interesting connection between the pixels chosen for perturbation by our approach and the *saliency map* of an image, as defined by Simonyan et al. (2014), that ranks pixels based on their influence on the output score. In effect our approach identifies pixels with high saliency scores but without explicitly using any gradient information (as needed in the definition of saliency map (Simonyan et al., 2014)). Intuitively, in each round, our local-search based approach computes an implicit approximation to the gradient of the current image by understanding the influence of a few pixels on the output, which is then used to update the current image.

**(3)** We perform extensive experimental evaluations, and show that our local-search based approach reliably generates adversarial examples with little perturbation (even when compared to a recent elegant adversarial attack proposed by Goodfellow et al. (2015) which needs perfect knowledge of the network). Another feature of our attack is that, by design, our approach only perturbs a very small fraction of the pixels during the adversarial image generation process (e.g., on the ImageNet1000 dataset we on average perturb only about $0.5\%$ of the pixels per image). Most previous attacks require the ability to perturb all the pixels in the image.

**(4)** Our approaches naturally extend to a stronger notion of misclassification (that we refer to as *$k$-misclassification*), where the goal is to ensure that the true label of the image does not even appear in the top-$k$ predictions of the network (obtained by sorting the confidence score vector). This notion especially captures the fact that many modern systems (e.g., ImageNet competition entrants) are evaluated based on top-$k$ predictions. To the best of our knowledge, these are the first adversarial attacks on deep neural networks achieving $k$-misclassification.

## 2 RELATED WORK

Starting with the seminal paper by Szegedy et al. (2014), which showed that the state-of-the-art neural networks are vulnerable to adversarial attacks, there has been significant attention focused on this problem. The research has led to investigation of different adversarial threat models and scenarios (Papernot et al., 2016c;b; Grosse et al., 2016; Kurakin et al., 2016; Fawzi et al., 2016), computationally efficient attacks (Goodfellow et al., 2015), perturbation efficient attacks (Moosavi-Dezfooli et al., 2016), etc.

Szegedy et al. (2014) used a box-constrained L-BFGS technique to generate adversarial examples. They also showed a transferability (or generalization) property for adversarial examples, in that adversarial examples generated for one network might also be misclassified by a related network with possibly different hyper-parameters (number of layers, initial weights, etc.). However, the need for a solving a series of costly penalized optimization problems makes this technique computationally expensive for generating adversarial examples. This issue was fixed by Goodfellow et al. (2015) who motivated by the underlying linearity of the components used to build a network proposed an elegant scheme based on adding perturbation proportional to sign of the network's cost function gradient. Recently, Moosavi-Dezfooli et al. (2016) used an iterative linearization procedure to generate adversarial examples with lesser perturbation. Another recent attack proposed by Papernot et al. (2016c) uses a notion of *adversarial saliency maps* (based on the saliency maps introduced by (Simonyan et al., 2014)) to select the most sensitive input components for perturbation. This attack has been adapted by Grosse et al. (2016) for generating adversarial samples for neural networks used as malware classifiers. However, all these above described attacks require perfect knowledge of the target network's architecture and parameters which limits their applicability to strong adversaries with the capability of gaining insider knowledge of the target system.

Our focus in this paper is the setting of black-box attacks, where we assume that an adversary has only the ability to use the network as an oracle. The adversary can obtain output from supplied inputs, and use the observed input-output relationship to craft adversarial images.[1] In the context of deep neural networks, a black-box attack was first proposed by Papernot et al. (2016b) with the motivation of constructing an attack on a remotely hosted system.[2] Their general idea is to first approximate the target network by querying it for output labels, which is used to train a substitute network, which is then used to craft adversarial examples for the original network. The success of the attack crucially depends on the transferability property to hold between the original and the substitute network. Our black-box attack is more direct, and completely avoids the transferability assumption, making it far more applicable. We also avoid the overhead of gathering data and training a substitute network. Additionally, our techniques can be adapted to a stronger notion of misclassification.

A complementary line of work has focused on building defenses against adversarial attacks. Although designing defenses is beyond scope of this paper, it is possible that adapting the previous suggested defense solutions such as *Jacobian-based regularization* (Gu & Rigazio, 2015) and *distillation* (Papernot et al., 2016d) can reduce the efficacy of our proposed attacks. Moreover, the recently proposed technique of *differentially private training* (Abadi et al., 2016) can also prove beneficial here.

The study of adversarial instability have led to development of solutions that seeks to improve training to in return increase the robustness and classification performance of the network. In some case, adding adversarial examples to the training (*adversarial training*) set can act like a regularizer (Szegedy et al., 2014; Goodfellow et al., 2015; Moosavi-Dezfooli et al., 2016). The phenomenon of adversarial instability has also been theoretically investigated for certain families of classifiers under various models of (semi) random noise (Fawzi et al., 2015; 2016). However, as we discuss later, due to peculiar nature of adversarial images generated by our approaches, a simple adversarial training is only mildly effective in preventing future similar adversarial attacks.

The security of machine learning in settings distinct from deep neural networks is also an area of active research with various known attacks under different threat models. We refer the reader to a recent survey by McDaniel et al. (2016) and references therein.

## 3 PRELIMINARIES

**Notation and Normalization.**    We denote by $[n]$ the set $\{1, \ldots, n\}$. The dataset of images is partitioned into train and test (or validation) subsets. An element of a dataset is a pair $(I, c(I))$ for an image $I$ and a ground truth label $c(I)$ of this image. We assume that the class labels are drawn from the set $\{1, \ldots, C\}$, i.e., we have a set of $C \in \mathbb{N}$ possible labels. We assume that images have $\ell$ channels (in experiments we use the RGB format) and are of width $w \in \mathbb{N}$ and height $h \in \mathbb{N}$. We say that $(b, x, y)$ is a coordinate of an image for channel $b$ and location $(x, y)$, and $(\star, x, y)$ is a pixel of an image where $(\star, x, y)$ represents all the $\ell$ coordinates corresponding to different channels at location $(x, y)$. $I(b, x, y) \in \mathbb{R}$ is the value of $I$ at the $(b, x, y)$ coordinate, and similarly $I(\star, x, y) \in \mathbb{R}^\ell$ represents the vector of values of $I$ at the $(\star, x, y)$ pixel.

It is a common practice to normalize the image before passing it to the network. A normalized image has the same dimension as the original image, but differs in the coordinate values. In this work we treat the normalization procedure as an external procedure and assume that all images are normalized. As we always work with normalized images, in the following, a reference to image means a normalized input image. We denote by LB and UB two constants such that all the coordinates of all the normalized images fall in the range [LB, UB]. Generally, LB < 0 and UB > 0. We denote by $\mathbb{I} \subset \mathbb{R}^{\ell \times w \times h}$ the space of all (valid) images which satisfy the following property: for every $I \in \mathbb{I}$, for all coordinates $(b, x, y) \in [\ell] \times [w] \times [h]$, $I(b, x, y) \in [\mathrm{LB}, \mathrm{UB}]$.

We denote by NN a trained neural network (trained on some set of training images). NN takes an image $I$ as an input and outputs a vector $\mathrm{NN}(I) = (o_1, \ldots, o_C)$, where $o_j$ denotes the probability as determined by NN that image $I$ belongs to class $j$. We denote $\pi(\mathrm{NN}(I), k)$ a function that returns a set of indices that are the top-$k$ predictions (ranked by decreasing probability scores with

---

[1]These kind of attacks are also known as *differential attacks* motivated by the use of the term in *differential cryptanalysis* (Biham & Shamir, 1991).
[2]Papernot et al. (2016a) have recently extended this attack beyond deep neural networks to other classes of machine learning techniques.

ties broken arbitrarily) of the network NN. For example, if $NN(I) = (0.25, 0.1, 0.2, 0.45)$, then $\pi(NN(I), 1) = \{4\}$ (corresponding to the location of the entry $0.45$). Similarly, $\pi(NN(I), 2) = \{4, 1\}, \pi(NN(I), 3) = \{4, 1, 3\}$, etc.

**Adversarial Goal.**    Before we define the goal of black-box adversarial attacks, we define misclassification for a NN. In this paper, we use a stronger notion of misclassification, which we refer to as $k$-misclassification for $k \in \mathbb{N}$.

**Definition 1 ($k$-misclassification)** *A neural network* NN $k$-misclassifies *an image $I$ with true label $c(I)$ iff the output $NN(I)$ of the network satisfies $c(I) \notin \pi(NN(I), k)$.*

In other words, $k$-misclassification means that the network ranks the true label below at least $k$ other labels. Traditionally the literature on adversarial attacks have only considered the case where $k = 1$. Note that an adversary that achieves a $k$-misclassification for $k > 1$ is a stronger adversary than one achieving an 1-misclassification ($k$-misclassification implies $k'$-misclassification for all $1 \leq k' \leq k$). If $k = 1$, we simply say that NN misclassifies the image.

In our setting, an adversary ADV is a function that takes in image $I$ as input and whose output is another image $ADV(I)$ (with same number of coordinates as $I$). We define an adversarial image as one that *fools* a network into $k$-misclassification.

**Definition 2 (Adversarial Image)** *Given access to an image $I$, we say that an $ADV(I)$ is a $k$-adversarial image (resp. adversarial image) if $c(I) \in \pi(NN(I), k)$ and $c(I) \notin \pi(NN(ADV(I)), k)$ (resp. $c(I) \in \pi(NN(I), 1)$ and $c(I) \notin \pi(NN(ADV(I)), 1)$).*

The goal of adversarial attacks is to design this function ADV that succeeds in fooling the network for a large set of images. Ideally, we would like to achieve this misclassification[3] by adding only some small perturbation (under some metric) to the image. The presence of adversarial images shows that there exist small perturbations in input that produce large perturbations at the output of the last layer.

Adversarial threat models can be divided into two broad classes[4]. The first class of models roughly assumes that the adversary has a total knowledge of the network architecture and the parameters resulting from training (or access to the labeled training set). The second class of threat models, as considered in this paper, make *no* assumptions about the adversary having access to the network architecture, network parameters, or the training set. In this case, the adversary has only a black-box (oracle) access to the network, in that it can query the network NN on an image $I$ and observe the output $NN(I)$. In our experimental section (Section 6), we also consider a slight weakening of this black-box model where the adversary has only the ability to use a proxy of the network NN as an oracle.

A black-box threat model in the context of deep neural networks was first considered by Papernot et al. (2016b). There is however one subtle difference between the threat model considered here and that considered by Papernot et al. (2016b) in what the adversary can access as an output. While the adversary presented in (Papernot et al., 2016b) requires access to the class label assigned by the network which is the same level of access needed by our simple randomized adversary (presented in Section 4), our local-search adversary (presented in Section 5) requires access to $o_{c(I)}$ (the probability assigned to the true label $c(I)$ by the network on input $I$) and the $\pi$ vector (for checking whether $k$-misclassification has been achieved). Our adversarial approaches does not require access to the complete probability vector ($NN(I)$). Also as pointed out earlier, compared to (Papernot et al., 2016b), our approach is more direct (needs no transferability assumption), requires no retraining, and can be adapted to achieve $k$-misclassification rather than just 1-misclassification.

## 4    BLACK-BOX GENERATION: A FIRST ATTEMPT

In this section, we present a simple black-box adversary that operates by perturbing a single pixel (or a small set of pixels) selected at random. In the next section, we build upon this idea to construct an adversary that achieves better success by making adaptive choices.

---

[3]Note that the misclassification is at test time, once the trained network has been deployed.

[4]More fine-grained classification has also been considered in (Papernot et al., 2016c) where adversaries are categorized by the information and capabilities at their disposal.

**Power of One Pixel.**    Starting point of our investigation is to understand the influence of a single pixel in an adversarial setting. Most existing adversarial attacks operate by applying the same perturbation on each individual pixel while minimizing the overall perturbation (Szegedy et al., 2014; Goodfellow et al., 2015; Moosavi-Dezfooli et al., 2016), while recent research have yielded attacks that perturb only a fraction of the pixels (Papernot et al., 2016c;b; Grosse et al., 2016). However, in all these cases, no explicit restriction is placed on the number of pixels that can be perturbed. Therefore, it is natural to ask: whether it is possible to force the network to misclassify an image by modifying a single pixel? If so, how strong should this perturbation be? We run several experiments to shed light on these questions. For simplicity, in this section, we focus the case of 1-misclassification, even though all discussions easily extend to the case of $k$-misclassification for $k > 1$. We begin with a useful definition.

**Definition 3 (Critical Pixel)** [5] *Given a trained neural network* NN *and an image $I$, a pixel $(\star, x, y)$ in $I$ is a critical pixel if a perturbation of this pixel generates an image that is misclassified by the network* NN. *In other words, $(\star, x, y)$ is a critical pixel in $I$ if there exists another neighboring image $I_p$ which differs from $I$ only in values at the pixel location $(x, y)$ such that $c(I) \notin \pi(\mathrm{NN}(I_p), 1)$.*

The image $I_p$ can be generated in multiple ways, here we consider a class of sign-preserving perturbation functions defined as follows. Let $\mathrm{PERT}(I, p, x, y)$ be a function that takes as input an image $I$, a perturbation parameter $p \in \mathbb{R}$, and a location $(x, y)$, and outputs an image $I_p^{(x,y)} \in \mathbb{R}^{\ell \times w \times h}$, defined as:

$$I_p^{(x,y)}(b, u, v) \stackrel{\mathrm{defn}}{=} \begin{cases} I(b, u, v) & \text{if } x \neq u \text{ or } y \neq v \\ p \times \mathrm{sign}(I(b, u, v)) & \text{otherwise} \end{cases} \tag{1}$$

In other words, the image $I_p^{(x,y)} = \mathrm{PERT}(I, p, x, y)$ has same values as image $I$ at all pixels except the pixel $(\star, x, y)$. The value of the image $I_p^{(x,y)}$ at pixel $(\star, x, y)$ is just $p \times \mathrm{sign}(I(\star, x, y))$. Note that unlike some of the previous adversarial attacks (Szegedy et al., 2014; Goodfellow et al., 2015; Moosavi-Dezfooli et al., 2016; Papernot et al., 2016c; Grosse et al., 2016), our threat model does not assume access to the true network gradient factors, hence the construction of the perturbed image has to be oblivious to the network parameters.

In the following, we say a pixel $(\star, x, y)$ in image $I$ is critical iff $c(I) \notin \pi(\mathrm{NN}(I_p^{(x,y)}), 1)$.

**Critical Pixels are Common.**    Our first experiment is to investigate existence of critical pixels in the considered dataset of images. To do so, we perform a simple procedure that picks a location $(x, y)$ in the image $I$ and applies the $\mathrm{PERT}$ function to this pixel to obtain a perturbed image $I_p^{(x,y)}$. Then the perturbed image is run through the trained network, and we check whether it was misclassified or not. If the perturbed image $I_p^{(x,y)}$ is misclassified then we have identified a critical pixel. While we can exhaustively repeat this procedure for all pixels in an image, for computational efficiency we instead perform it only on a fraction of randomly chosen pixels, and our results somewhat surprisingly suggest that in many cases this is sufficient to generate an adversarial image. Algorithm RANDADV presents the pseudo-code for this experiment. Algorithm RANDADV, selects $U$ random pixels (with replacement) and performs checks whether the pixel is critical or not. The algorithm output is an unbiased estimate for the fraction of critical pixels in the input image $I$. Note that the algorithm can fail in generating an adversarial image (i.e., in finding any critical pixel for an image). The following definition will be useful for our ensuing discussion.

**Definition 4 (Good Image)** *We say that an image $I$ with true label $c(I)$ is* good *for a network* NN *iff $c(I) \in \pi(NN(I), 1)$ (i.e.,* NN *predicts $c(I)$ as the most likely label for $I$).*

Our first observation is that sometimes even small perturbation to a pixel can be sufficient to obtain an adversarial image. Table 2 shows two images and their adversarial counterparts, with $p = 1$. Often, original and adversarial images are indistinguishable to the human eye, but sometimes the critical pixel is visible (Table 2).

---

[5]In the definition of critical pixel we have not considered how well the original image $I$ is classified by NN, i.e., whether $c(I) \in \pi(\mathrm{NN}(I), 1)$. In particular, if $c(I) \notin \pi(\mathrm{NN}(I), 1)$ then by definition all pixels in the image are critical even without any perturbation. In our experiments, we ignore these images and only focus on images $I$ where $c(I) \in \pi(\mathrm{NN}(I), 1)$, which we refer to as *good* images (Definition 4).

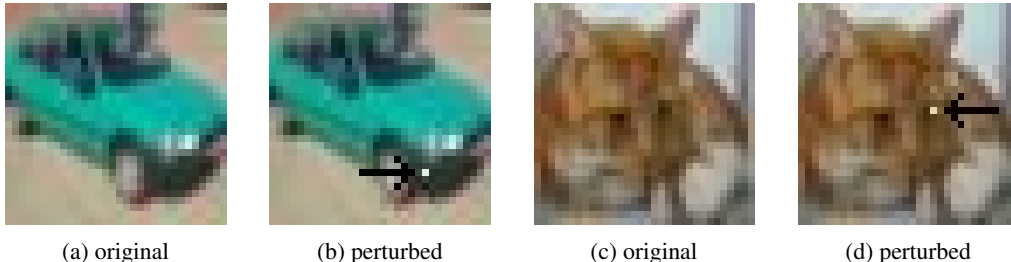

| (a) original | (b) perturbed | (c) original | (d) perturbed |

Table 2: The row contains original images followed by misclassified images where only one pixel (pointed using a black arrow) was perturbed with perturbation parameter $p = 1$. After perturbation, in the first case (images (a) and (b)) an automobile gets misclassified as a truck, and in the second case (images (c) and (d)) a cat gets misclassified as a dog.

We also tried to understand the effect of larger perturbation parameter values. We set $U$ to half the number of pixels in each image. After usual training of the neural network using the training set (see Section 6 for more details about training), we ran Algorithm RANDADV on 1000 randomly drawn images from the test set of the corresponding dataset. In our experiments, we varied perturbation parameter in the range $\{1, 5, 10, 100\}$. Before we consider our results, we note some of the perturbation values that we use to construct the adversarial image might construct images that are not in the original image space.[6] However, these results are still somewhat surprising, because even though we allow large (even out-of-range) perturbation, it is applied to exactly *one pixel* in the image, and it appears that it suffices to even pick the pixel at random.

Figures 1 and 2 show results for 4 datasets (more details about the datasets and the networks are presented in Section 6). On the x-axis we show the perturbation parameter $p$. In Figure 1, the y-axis represents the output of Algorithm RANDADV averaged over good images for the network.[7] The first observation that we can make is that the critical pixels are common, and in fact, as $p$ grows the fraction of critical pixels increases. For example, in CIFAR10, with $p = 100$, almost 80% (on average) of the pixels randomly selected are critical. In Figure 2, the y-axis represents the fraction of successful adversarial images generated by Algorithm RANDADV, i.e., fraction of inputs where Algorithm RANDADV is successful in finding at least one critical pixel. Again we notice that as $p$ grows it gets easier for Algorithm RANDADV to construct an adversarial image.

---

**Algorithm 1** RANDADV (NN)

---

1: **Input:** Image $I$ with true label $c(I) \in \{1, \ldots, C\}$, perturbation factor $p \in \mathbb{R}$, and a budget $U \in \mathbb{N}$ on the number of trials
2: **Output:** A randomized estimate on the fraction of critical pixels in the input image $I$
3: $i = 1$, critical $= 0$
4: **while** $i \leq U$ **do**
5: randomly pick a pixel $(\star, x, y)$
6: compute a perturbed image $I_p^{(x,y)} = \text{PERT}(I, p, x, y)$
7: **if** $c(I) \notin \pi(\text{NN}(I_p^{(x,y)}), 1)$ **then**
8: critical $\leftarrow$ critical $+ 1$
9: **end if**
10: $i \leftarrow i + 1$
11: **end while**
12: {The algorithm succeeds in generating an adversarial image if it finds at least one critical pixel}
13: **return** $\frac{\text{critical}}{U}$

---

Another observation is that for the MNIST and STL10 datasets, Algorithm RANDADV succeeds in finding fewer critical pixels as compared to SVHN and CIFAR10 datasets. We give the following explanation for this observation. The majority of pixels in an MNIST image belong to the background,

---

[6]We fix this shortcoming using a local-search based strategy in the next section.
[7]Note by focusing on good images, we make sure that we are only accounting for those cases where perturbation is needed for creating an adversarial image.

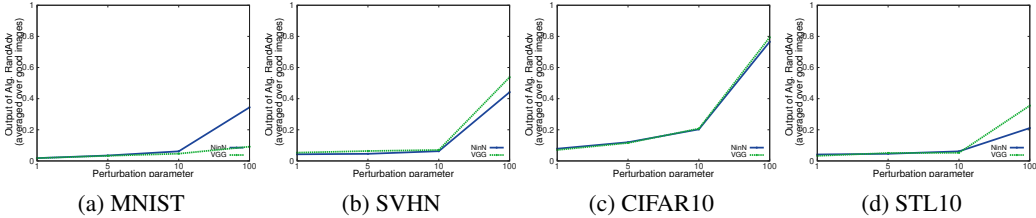

Figure 1: Output of Algorithm RANDADV (averaged over good images). The results are for two networks: a) Network-in-Network and b) VGG. The perturbation parameter $p$ is varied from $\{1, 5, 10, 100\}$.

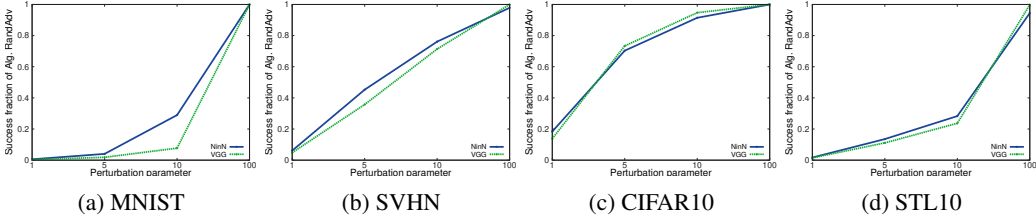

Figure 2: Fraction of images where Algorithm RANDADV succeeds in finding at least one critical pixel. Again we only start with only good images.

hence, these pixels are less likely to be critical. On the other hand, STL10 contains high resolution images, $96 \times 96$, where perhaps a single pixel has less of an impact on the output prediction. The latter observation motivated us to generalize the notion of a critical pixel to a *critical set*.

**Definition 5 (Critical Set)** *Given a trained neural network* NN *and an image $I$, a critical set of $I$ is a set of pixels $\bigcup_{(x,y)}\{(\star, x, y)\}$ in $I$ such that a perturbation of these pixels generates an image that is misclassified by the network* NN.

The general goal will be to find critical sets of small size in an image. With this notion of critical set, we considered constructing adversarial images on the high-resolution ImageNet1000 dataset. We can modify the definition of $I_p^{(x,y)}$ (from (1)) where instead of a single pixel we perturb all the pixels in a set. Similarly, we can devise a simple extension to Algorithm RANDADV to operate with a set of pixels and to output an unbiased estimate for the fraction of critical sets of some fixed size (50 in our case) in the input image.[8] Note that a set size of 50 pixels is still a tiny fraction of all the pixels in a standard (center) crop of size $224 \times 224$, namely just $0.09\%$. We use a larger perturbation parameter $p$ than before, and set $(U)$ the budget on the number of trials on an image as 5000. Figure 3 shows our results. Overall, we note that we can draw similar conclusions as before, i.e., increasing the perturbation parameter creates more critical sets making them easier to find and relatively small perturbations are sufficient to construct adversarial images.

## 5 BLACK-BOX GENERATION: A GREEDY APPROACH

The results from Section 4 show that most images have critical pixels such that modifying these pixels significantly leads to a failure of NN to classify the image correctly. However, one shortcoming of Algorithm RANDADV was that to build adversarial images, we sometimes had to apply a large perturbation to a single pixel (or a small set of pixels). Hence, there might exist a pixel (or a set of pixels) in the adversarial image whose coordinate value could lie outside the valid range [LB, UB]. To overcome this issue, we need to redesign the search procedure to generate adversarial images that still belong to the original image space $\mathbb{I}$ (defined in Section 3). Here a brute-force approach is generally not feasible because of computational reasons, especially in high-resolution images. Hence, we need to develop an efficient heuristic procedure to find the right small set of pixels to be perturbed. Our solution presented in this section is based on performing a greedy local search over the image space.

---

[8]Searching over all pixel sets of size 50 pixels is computationally prohibitive, which again motivates the need for a randomized strategy as proposed in Algorithm RANDADV.

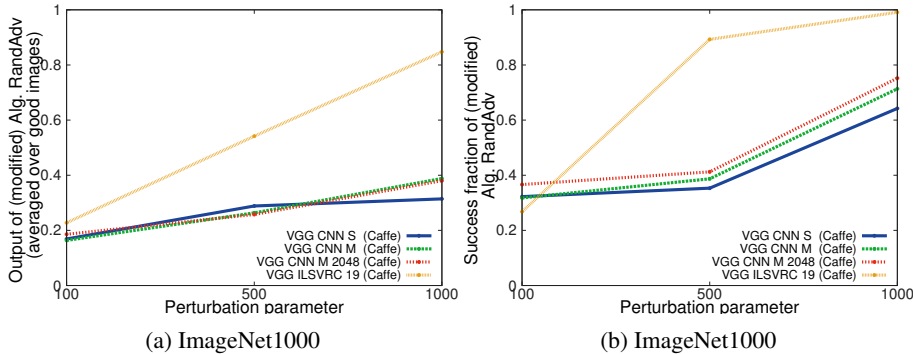

Figure 3: Experiments in Figures 1 and 2 for the high-resolution ImageNet1000 dataset. The results are again for good images from a set of 1000 randomly selected images. We use a slightly modified version of Algorithm RANDADV that perturbs a set of 50 pixels.

We consider the general $k$-misclassification problem (Definition 1) where an adversarial attack ensures that the true label does not appear in the top-$k$ predictions of the network. We utilize a local-search procedure, which is an incomplete search procedure that is widely used for solving combinatorial problems appearing in diverse domains such as graph clustering, scheduling, logistics, and verification (Lenstra, 1997). For a general optimization problem it works as follows. Consider an objective function $f(\mathbf{z}) : \mathbb{R}^n \to \mathbb{R}$ where the goal is to minimize $f(\mathbf{z})$. The local-search procedure works in rounds, where each round consists of two steps. Let $\mathbf{z}_{i-1}$ be the solution iterate after round $i - 1$. Consider round $i$. The first step is to select a small subset of points $Z = \{\hat{\mathbf{z}}_1, \ldots, \hat{\mathbf{z}}_n\}$, a so called *local neighborhood*, and evaluate $f(\hat{\mathbf{z}}_j)$ for every $\hat{\mathbf{z}}_j \in Z$. Usually, the set $Z$ consist of points that are close to current $\mathbf{z}_{i-1}$ for some measure of distance which is domain specific. The second step selects a new solution $\mathbf{z}_i$ taking into account the previous solution $\mathbf{z}_{i-1}$ and the points in $Z$. Hence, $\mathbf{z}_i = g(f(\mathbf{z}_{i-1}), f(\hat{\mathbf{z}}_1), \ldots, f(\hat{\mathbf{z}}_n))$, where $g$ is some pre-defined *transformation function*.

We adapt this general procedure to search critical sets efficiently as explained below. Our optimization problem will try to minimize the probability that the network determines an perturbed image has the class label of the original image, and by using a local-search procedure we generate perturbed images which differ from the original image in only few pixels. Intuitively, in each round, our local-search procedure computes an implicit approximation to the gradient of the current image by understanding the influence of a few pixels on the output, which is then used to update the current image.

**(a)** First, we need to define the cost function $f$. Let $I$ be the image (with true label $c(I)$) whose adversarial image we want to generate for a target neural network NN. For some input image $\hat{I}$, we use the objective function $f_{c(I)}(\hat{I})$ which equals the probability assigned by the network NN that the input image $\hat{I}$ belongs to class $c(I)$. More formally,

$$f_{c(I)}(\hat{I}) = o_{c(I)} \text{ where } \text{NN}(\hat{I}) = (o_1, \ldots, o_C),$$

with $o_j$ denoting the probability as determined by NN that image $\hat{I}$ belongs to class $j$. Our local-search procedure aims to minimize this function.

**(b)** Second, we consider how to form a neighborhood set of images. As mentioned above, the local-search procedure operates in rounds. Let $\hat{I}_{i-1}$ be the image after round $i - 1$. Our neighborhood will consist of images that are different in one pixel from the image $\hat{I}_{i-1}$. In other words, if we measure the distance between $\hat{I}_{i-1}$ and any image in the neighborhood as the number of perturbed pixels, then this distance is the same (equal to one) for all of them. Therefore, we can define the neighborhood in terms of a set of pixel locations. Let $(P_X, P_Y)_i$ be a set of pixel locations. For the first round $(P_X, P_Y)_0$ is randomly generated. At each subsequent round, it is formed based on a set of pixel locations which were perturbed in the previous round. Let $(P_X^*, P_Y^*)_{i-1}$ denote the pixel locations that were perturbed in round $i - 1$ (formally defined below). Then

$$(P_X, P_Y)_i = \bigcup_{\{(a,b) \in (P_X^*, P_Y^*)_{i-1}\}} \bigcup_{\{x \in [a-d, a+d], y \in [b-d, b+d]\}} (x, y),$$

where $d$ is a parameter. In other words, we consider pixels that were perturbed in the previous round, and for each such pixel we consider all pixels in a small square with the side length $2d$ centered at that pixel. This defines the neighborhood considered in round $i$.

(c) Third, we describe the transformation function $g$ of a set of pixel locations. The function $g$ takes as input an image $\hat{I}$, a set of pixel locations $(P_X, P_Y)$, a parameter $t$ that defines how many pixels will be perturbed by $g$, and two perturbation parameters $p$ and $r$. In round $i$ of the local-search procedure, the function $g(\hat{I}_{i-1}, (P_X, P_Y)_{i-1}, t, p, r)$ outputs a new image, such that exactly $t$ pixels of $\hat{I}_{i-1}$ are perturbed, and an auxiliary set of pixel locations $(P_X^*, P_Y^*)_i$ to record which $t$ pixels where perturbed at this round, so we have $(\hat{I}_i, (P_X^*, P_Y^*)_i) = g(\hat{I}_{i-1}, (P_X, P_Y)_{i-1}, t, p, r)$. Next we describe transformations that $g$ performs in round $i$. As the first step, $g$ constructs a set of perturbed images based on $(P_X, P_Y)_{i-1}$:

$$\mathcal{I} = \bigcup_{(x,y) \in (P_X, P_Y)_{i-1}} \{\text{PERT}(\hat{I}_{i-1}, p, (x, y))\},$$

where PERT is the perturbation function defined through (1). Then it computes the score of each image in $\mathcal{I}$ as

$$\forall \tilde{I} \in \mathcal{I} \; : \; \text{score}(\tilde{I}) = f_{c(I)}(\tilde{I}),$$

and it sorts (in decreasing order) images in $\mathcal{I}$ based on the above score function to construct sorted($\mathcal{I}$). Pixels whose perturbation lead to a larger decrease of $f$ are more likely useful in constructing an adversarial candidate. From sorted($\mathcal{I}$), it records a set of pixel locations $(P_X^*, P_Y^*)_i$ based on the first $t$ elements of sorted($\mathcal{I}$), where the parameter $t$ regulates the number of pixels perturbed in each round. Formally,

$$(P_X^*, P_Y^*)_i = \{(x, y) \; : \; \text{PERT}(\hat{I}_{i-1}, p, (x, y)) \in \text{sorted}(\mathcal{I})[1:t]\},$$

where sorted($\mathcal{I}$)$[1:t]$ represents the first $t$ sorted images in sorted($\mathcal{I}$). Finally, $\hat{I}_i$ is constructed from $\hat{I}_{i-1}$ by perturbing each pixel in location $(x, y) \in (P_X^*, P_Y^*)_i$ with a perturbation value $r$. The perturbation is performed in a *cyclic* way (as explained in Algorithm CYCLIC) so that we make sure that all coordinate values in $\hat{I}_i$ are within the valid bounds of LB and UB. Note that at the end of every round $i$, $\hat{I}_i$ is a valid image from the image space $\mathbb{I}$.

We want to point out that the function $g$ uses two perturbation parameters, $p$ and $r$. The value of $r$ is kept small in the range $[0, 2]$. On the other hand, we do not put any explicit restrictions on the value of $p$. The best choice of $p$ will be one that facilitates the identification of the "best" pixels to perturb in each round. In our experiments, we adjust the value of $p$ automatically during the search. We defer this discussion to the experimental section.

Algorithm LOCSEARCHADV shows the complete pseudocode of our local-search procedure. At the high level, the algorithm takes an image as input, and in each round, finds some pixel locations to perturb using the above defined objective function and then applies the above defined transformation function to these selected pixels to construct a new (perturbed) image. It terminates if it succeeds to push the true label below the $k$th place in the confidence score vector at any round. Otherwise, it proceeds to the next round (for a maximum of $R$ rounds). Note that the number of pixels in an image perturbed by Algorithm LOCSEARCHADV is at most $t \times R$ and in practice (see Tables 4, 5,and 6 in Section 6) it is much less. In round $i$, we query the network at most the number of times as the number of pixels in $(P_X, P_Y)_i$ which after the first round is at most $2d \times 2d \times t$ (again in practice this is much less because of the overlaps in the neighborhood squares).

In Section 6, we demonstrate the efficacy of Algorithm LOCSEARCHADV in constructing adversarial images. We first highlight an interesting connection between the pixels perturbed and their influences measured by a notion of called saliency map.

**A Relation to Saliency Maps.** Simonyan et al. (2014) introduced the notion of saliency map as a way to rank pixels of the original images w.r.t. their influence on the output of the network. The intuition is that influential pixels in the saliency map are more likely to be important pixels that represent objects and, for example, can be used for weakly supervised object localization. Formally, let $\text{NN}_{c(I)}(I)$ denote the probability assigned to true class $c(I)$ by the network NN on input $I \in \mathbb{R}^{\ell \times w \times h}$. Let $W_{c(I)} \in R^{\ell \times w \times h}$ denote the derivative of $\text{NN}_{c(I)}$ with respect to the input evaluated at image $I$.

---

**Algorithm 2** CYCLIC $(r, b, x, y)$

---

**Assumptions:** Perturbation parameter $r \in [0, 2]$ and $\text{LB} \leq 0 \leq \text{UB}$
**Output:** Perturbed image value at the coordinate $(b, x, y)$ which lies in the range $[\text{LB}, \text{UB}]$

**if** $rI(b, x, y) < \text{LB}$ **then**
 **return** $rI(b, x, y) + (\text{UB} - \text{LB})$
**else if** $rI(b, x, y) > \text{UB}$ **then**
 **return** $rI(b, x, y) - (\text{UB} - \text{LB})$
**else**
 **return** $rI(b, x, y)$
**end if**

---

**Algorithm 3** LOCSEARCHADV (NN)

---

**Input:** Image $I$ with true label $c(I) \in \{1, \ldots, C\}$, two perturbation parameters $p \in \mathbb{R}$ and $r \in [0, 2]$, and four other parameters: the half side length of the neighborhood square $d \in \mathbb{N}$, the number of pixels perturbed at each round $t \in \mathbb{N}$, the threshold $k \in \mathbb{N}$ for $k$-misclassification, and an upper bound on the number of rounds $R \in \mathbb{N}$.
**Output:** Success/Failure depending on whether the algorithm finds an adversarial image or not

$\hat{I}_0 = I, i = 1$
Pick 10% of pixel locations from $I$ at random to form $(P_X, P_Y)_0$
**while** $i \leq R$ **do**
 {Computing the function $g$ using the neighborhood}
 $\mathcal{I} \leftarrow \bigcup_{(x,y) \in (P_X, P_Y)_{i-1}} \{\text{PERT}(\hat{I}_{i-1}, p, x, y)\}$
 Compute $\text{score}(\tilde{I}) = f_{c(I)}(\tilde{I})$ for each $\tilde{I} \in \mathcal{I}$ (where $f_{c(I)}(\tilde{I}) = o_{c(I)}$ with $\text{NN}(\tilde{I}) = (o_1, \ldots, o_C)$)
 $\text{sorted}(\mathcal{I}) \leftarrow$ images in $\mathcal{I}$ sorted by descending order of score
 $(P_X^*, P_Y^*)_i \leftarrow \{(x, y) : \text{PERT}(\hat{I}_{i-1}, p, x, y) \in \text{sorted}(\mathcal{I})[1 : t]\}$ (with ties broken arbitrarily)
 {Generation of the perturbed image $\hat{I}_i$}
 **for** $(x, y) \in (P_X^*, P_Y^*)_i$ and each channel $b$ **do**
 $\hat{I}_i(b, x, y) \leftarrow$ CYCLIC $(r, b, x, y)$
 **end for**
 {Check whether the perturbed image $\hat{I}_i$ is an adversarial image}
 **if** $c(I) \notin \pi(\text{NN}(\hat{I}_i), k)$ **then**
 **return** Success
 **end if**
 {Update a neighborhood of pixel locations for the next round}
 $(P_X, P_Y)_i \leftarrow \bigcup_{\{(a,b) \in (P_X^*, P_Y^*)_{i-1}\}} \bigcup_{\{x \in [a-d, a+d], y \in [b-d, b+d]\}} (x, y)$
 $i \leftarrow i + 1$
**end while**
**return** Failure

---

The saliency map of $I$ is the matrix $M \in \mathbb{R}^{w \times h}$ such that $M_{i,j} = \max_{b \in [\ell]} |W_{c(I)}(b, x, y)|$, where $W_{c(I)}(b, x, y)$ is the element of $W_{c(I)}$ corresponding to channel $b$ and location $(x, y)$. Pixels with higher scores are considered more influential. In subsequent works, this notion has been extended to adversarial saliency maps that can be useful in generating adversarial perturbations (Papernot et al., 2016c).

Computing the exact saliency scores for an image requires complete access to the network NN, which we do not assume. However, a natural hypothesis is that the pixels selected by Algorithm LOC-SEARCHADV for perturbation are related to pixels with large saliency scores. We use the ImageNet1000 dataset to test this hypothesis. In Figure 3, we present some qualitative results. As can be seen from the pictures, the pixels perturbed by Algorithm LOCSEARCHADV appear correlated with pixels with high saliency scores. Quantitatively, we observed that the pixels that occupy top-10% of the saliency map, on average contain more than 23% of the pixels chosen by Algorithm LOC-SEARCHADV for perturbation (and this overlap only grows when we consider a bigger chunk of pixels picked by their saliency scores). Note that this is correlation is not though a random occurrence. For an image $I$, let $S_I$ denote the set of pixels in $I$ that rank among the top-10% in the saliency map. If we pick a random set of around 200 pixels (this is on average number of pixels perturbed per image by Algorithm LOCSEARCHADV perturbs, see Table 5), we expect only about 10% to them to intersect with $S_I$ and standard tail bounds show that the probability that at least 23% of the pixels of this random set intersects with $S_I$ is extremely small.[9] Therefore, it appears that Algorithm LOCSEARCHADV rediscovers part of the high salient score pixels but without explicitly computing the gradients.

## 6    EXPERIMENTAL EVALUATION

We start by describing our experimental setup. We used Caffe and Torch machine learning frameworks to train the networks. All algorithms to generate adversarial images were implemented in Lua within Torch 7. All experiments were performed on a cluster of GPUs using a single GPU for each run.

**Datasets.**    We use 5 popular datasets: MNIST (handwritten digits recognition dataset), CIFAR10 (objects recognition dataset), SVHN (digits recognition dataset), STL10 (objects recognition dataset), and ImageNet1000 (objects recognition dataset).

**Models.**    We trained Network-in-Network (Lin et al., 2014) and VGG (Simonyan & Zisserman, 2014) for MNIST, CIFAR, SVHN, STL10, with minor adjustments for the corresponding image sizes. Network-in-Network is a building block of the commonly used GoogLeNet architecture that has demonstrated very good performance on medium size datasets, e.g. CIFAR10 (Zagoruyko, 2015). VGG is another powerful network that proved to be useful in many applications beyond image classification, like object localization (Ren et al., 2015). We trained each model in two variants: with and without batch normalization (Ioffe & Szegedy, 2015). Batch normalization was placed before a ReLU layer in all networks. For the ImageNet1000 dataset, we used pre-trained VGG models from (Chatfield et al., 2014b) (we did not train them from scratch due to limited resources). All Caffe VGG models were converted to Torch models using the *loadcaffe* package (Zagoruyko, 2016a). These models use different normalization procedures which we reproduced for each model based on provided descriptions. Tables 4 and 5 (the second column ERRTOP-1) show the top-1 (base) error for all datasets and models that we considered. The results are comparable with the known state-of-the-art results on these datasets (Benenson, 2016).

**Related Techniques.**    There are quite a few approaches for generating adversarial images (as discussed in Section 2). Most of these approaches require access to the network architecture and its parameter values (Szegedy et al., 2014; Goodfellow et al., 2015; Moosavi-Dezfooli et al., 2016; Papernot et al., 2016c).[10]  The general idea behind these attacks is based on the evaluating the network's sensitivity to the input components in order to determine a perturbation that achieves the adversarial misclassification goal. Among these approaches, the attack approach (known as

---

[9]We can also use here a standard hypothesis testing for a proportion. The null-hypothesis is that the probability of intersection equals 0.1 as with random Bernoulli trails, and test statistic $Z = (0.23 - 0.1)/\sqrt{(0.1)(1-0.1)/200} = 6.12$ indicates that the null-hypothesis can be rejected at significance level 0.01.

[10]Therefore, not entirely suited for a direct comparison with our black-box approach.

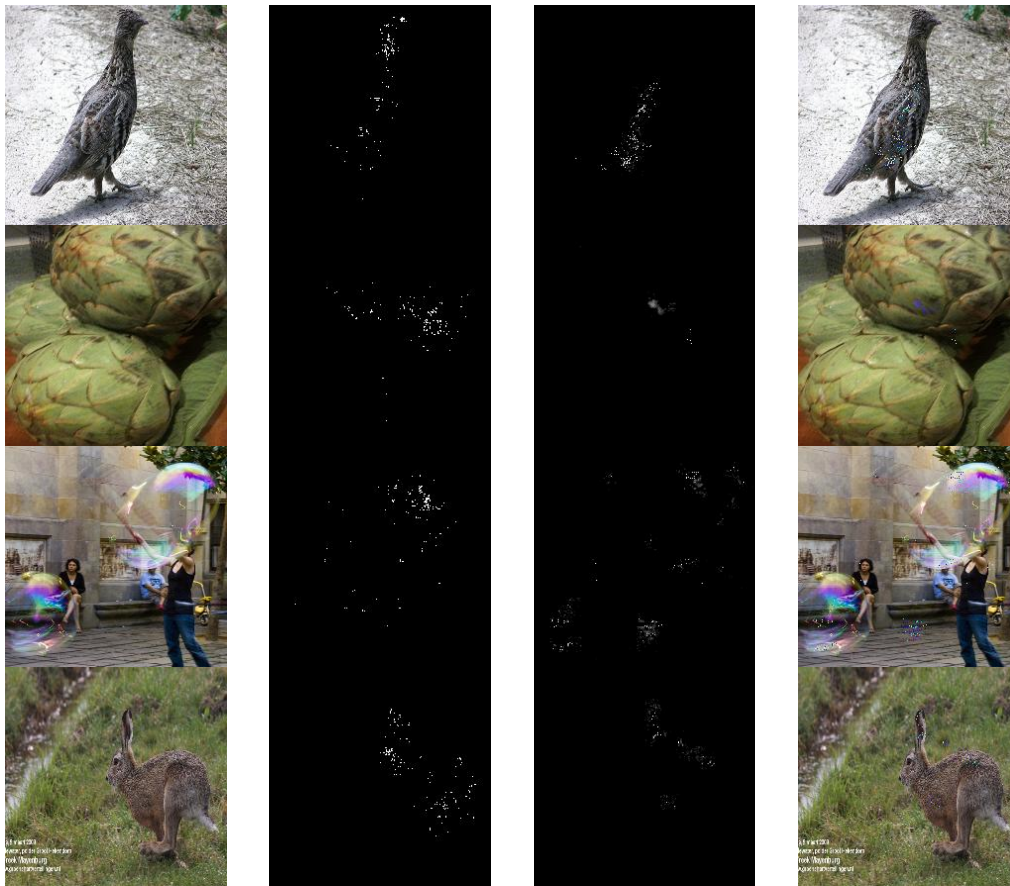

Table 3: Results on ImageNet1000 using VGG CNN-S (Caffe) network (Chatfield et al., 2014a). Columns from left to right: the original image, top 150 pixels chosen according to their saliency scores (in white), the absolute difference between the perturbed image and the true image (the pixels that are perturbed appear in white), and the perturbed image. Adversarial misclassification (rows from top to bottom): a ruffed grouse misclassified as a frilled lizard, an artichoke misclassified as a sleeping bag, a bubble misclassified as a fountain, and a hare misclassified as a cheetah.

the "fast-gradient sign method") suggested by Goodfellow et al. (2015) stands out for being able to efficiently generate adversarial images. Here we compare the performance of our local-search based attack against this fast-gradient sign method.[11]

For completeness, we now briefly explain the fast-gradient sign method of Goodfellow et al. (2015). Given an image $I_0$, a label $a \in \{1, \ldots, C\}$, and a network NN, the fast-gradient sign method perturbs $I_0$ using the following update rule: $I_0^{\text{pert}} = I_0 + \epsilon \cdot \text{sign}(\nabla_{I=I_0} \text{Loss}(\text{NN}(I), a))$ where $\text{sign}(\nabla_{I=I_0} \text{Loss}(\text{NN}(I), a))$ is the sign of the network's cost function gradient (here $\text{Loss}(\text{NN}(I), a)$ denotes the loss function of the network NN given input $I$ and class $a$). We vary $a$ over all possible labels in the dataset and choose the best result where this procedure is successful in generating an adversarial image. Without general guidelines for setting $\epsilon$, we experimented with several values of $\epsilon$ starting from $0.07$ and increasing this number. We found that the value $\epsilon = 0.2$[12] was the smallest value where the fast-gradient sign method started to yield competitive performance compared to our algorithm. Smaller values of $\epsilon$ leads to generation of fewer adversarial images, e.g., at $\epsilon = 0.1$, the percentage of generated adversarial images is reduced by around 10% as compared to the value at $\epsilon = 0.2$ for the CIFAR10 dataset on the Network-in-Network model. Larger values of $\epsilon$ tends to

---

[11]Another reason for picking this approach for comparison is that it is also heavily utilized in the recent black-box attack suggested by Papernot et al. (2016b), where they require additional transferability assumptions which is not required by our attack.

[12]For the ImageNet1000 dataset, we set $\epsilon$ differently as discussed later.

generate more adversarial images, but this comes at the cost of an increase in the perturbation. As we discuss later, our local-search based approach yields better results than the fast-gradient sign method in both the volume of adversarial images generated and the amount of perturbation applied. Another important point to remember is that unlike the fast-gradient sign method, our approach is based on a weaker and more realistic assumption on the adversarial power, making our attacks more widely applicable.

**Implementing Algorithm LOCSEARCHADV.** For each image $I$, we ran Algorithm LOC-SEARCHADV for at most 150 rounds, perturbing 5 pixels at each round, and use squares of side length 10 to form the neighborhood (i.e., $R = 150, t = 5, d = 5$). With this setting of parameters, we perturb a maximum of $t \times R = 750$ pixels in an image. The perturbation parameter $p$ was adaptively adjusted during the search. This helps in faster determination of the most helpful pixels in generating the adversarial image. Let $I$ be the original image. For some round $i$ of the algorithm, define $\bar{o}_{c(I)} = \mathrm{avg}_{(x,y)}\{o_{c(I)} : (x,y) \in (P_X^*, P_Y^*)_{i-1}\}$, where $o_{c(I)}$ is the probability assigned to class label $c(I)$ in $\mathrm{NN}(\mathrm{PERT}(\hat{I}_{i-1}, p, x, y))$ (here $\bar{o}_{c(I)}$ provides an approximation of the average confidence of the network NN in predicting the true label over perturbed images). At each round, we increase the value of $p$ if $\bar{o}_{c(I)}$ is close to one and decrease $p$ if $\bar{o}_{c(I)}$ is low, e.g., below 0.3. For Algorithm CYCLIC, we set $r = 3/2$. To avoid perturbing the most sensitive pixels frequently, we make sure that if a pixel is perturbed in a round then we exclude it from consideration for the next 30 rounds.

**Experimental Observations.** For ease of comparison with the fast-gradient sign method (Goodfellow et al., 2015), we set $k = 1$ and focus on achieving 1-misclassification. Tables 4 and 5 show the results of our experiments on the test sets. The first column shows the dataset name. The second column (ERRTOP-1) presents the top-1 misclassification rate on the corresponding test dataset without any perturbation (base error). ERRTOP-1(ADV) is the top-1 misclassification rate where each original image in the test set was replaced with an generated perturbed image (using either our approach or the fast-gradient sign method (Goodfellow et al., 2015) which is denoted as FGSM).[13]

In the following, we say an adversarial generation technique ADV, given an input image $I$, succeeds in generating an adversarial image $\mathrm{ADV}(I)$ for a network NN iff $c(I) \in \pi(\mathrm{NN}(I), 1)$ and $c(I) \notin \pi(\mathrm{NN}(\mathrm{ADV}(I)), 1)$. The CONF column shows the average confidence over all successful adversarial images for the corresponding technique. The PTB column shows the average (absolute) perturbation added per coordinate in cases of successful adversarial generation. More formally, let $\mathcal{T}$ denote the test set and $\mathcal{T}_{\mathrm{ADV}} \subseteq \mathcal{T}$ denote the set of images in $\mathcal{T}$ on which ADV is successful. Then,

$$\mathrm{PTB} = \frac{1}{|\mathcal{T}_{\mathrm{ADV}}|} \sum_{I \in \mathcal{T}_{\mathrm{ADV}}} \frac{1}{\ell \times w \times h} \sum_{b,x,y} |I(b,x,y) - \mathrm{ADV}(I)(b,x,y)|,$$

where $I \in \mathbb{R}^{\ell \times w \times h}$ is the original image and $\mathrm{ADV}(I) \in \mathbb{R}^{\ell \times w \times h}$ is the corresponding adversarial image. Note that the inner summation is measuring the $L_1$-distance between $I$ and $\mathrm{ADV}(I)$. The #PTBPIXELS column shows the average percentage of perturbed pixels in the successful adversarial images. Similarly, TIME column shows the average time (in seconds) to generate a successful adversarial image. Finally, the last column indicates the type of network architecture.

As is quite evident from these results, Algorithm LOCSEARCHADV is more effective than the fast-gradient sign method in generating adversarial images, even without having access to the network architecture and its parameter values. The difference is quite prominent for networks trained with batch normalization as here we noticed that the fast-gradient sign method has difficulties producing adversarial images.[14] Another advantage with our approach is that it modifies a very tiny fraction of pixels as compared to all the pixels perturbed by the fast-gradient sign method, and also in many cases with far less average perturbation. Putting these points together demonstrates that

---

[13]Note that by explicitly constraining the number of pixels that can be perturbed, as we do in our approach, it might be impossible to get to a 100% misclassification rate on some datasets. Similarly, the fast-gradient sign method fails to achieve a 100% misclassification rate even with larger values of $\epsilon$ (Moosavi-Dezfooli et al., 2016).

[14]In general, we observed that models trained with batch normalization are somewhat more resilient to adversarial perturbations probably because of the regularization properties of batch normalization (Ioffe & Szegedy, 2015).

Algorithm LOCSEARCHADV is successful in generating more adversarial images than the fast-gradient sign method, while modifying far fewer pixels and adding less noise per image. On the other side, the fast-gradient sign method takes lesser time in the generation process and generally seems to produce higher confidence scores for the adversarial (misclassified) images.

Table 5 shows the results for several variants of VGG network trained on the ImageNet1000 dataset. These networks do not have batch normalization layers (Chatfield et al., 2014b; Zagoruyko, 2016a). We set $\epsilon = 1$ for the fast-gradient sign method as a different pre-processing technique was used for this network (we converted these networks from pre-trained Caffe models). Results are similar to that observed on the smaller datasets. In most cases, our proposed local-search based approach is more successful in generating adversarial images while on average perturbing less than $0.55\%$ of the pixels.

**Case of Larger $k$'s.** We now consider achieving $k$-misclassification for $k \geq 1$ using Algorithm LOCSEARCHADV. In Table 6, we present the results as we change the goal from 1-misclassification to 4-misclassification on the CIFAR10 dataset. We use the same parameters as before for Algorithm LOCSEARCHADV. As one would expect, as we increase the value of $k$, the effectiveness of the attack decreases, perturbation and time needed increases. But overall our local-search procedure is still able to generate a large fraction of adversarial images at even $k = 4$ with a small perturbation and computation time, meaning that these images will fool even a system that is evaluated on a top-4 classification criteria. We are not aware of a straightforward extension of the fast-gradient sign method (Goodfellow et al., 2015) to achieve $k$-misclassification.

**Even Weaker Adversarial Models.** We also consider a weaker model where the adversary does not even have a black-box (oracle) access to the network (NN) of interest, and has to rely on a black-box access to somewhat of a "similar" (proxy) network as NN. For example, the adversary might want to evade a spam filter A, but might have to develop adversarial images by utilizing the output of a spam filter B, which might share properties similar to A.

We trained several modifications of Network-in-Network model for the CIFAR10 dataset, varying the initial value of the learning rate, the size of filters, and the number of layers in the network. We observed that between 25% to 43% of adversarial images generated by Algorithm LOCSEARCHADV using the original network were also adversarial for these modified networks (at $k = 1$). The transferability of adversarial images that we observe here has also been observed with other attacks too (Szegedy et al., 2014; Goodfellow et al., 2015; Papernot et al., 2016b;a) and demonstrates the wider applicability of all these attacks.

# 7 CONCLUSION

We investigate the inherent vulnerabilities in modern CNNs to practical black-box adversarial attacks. We present approaches that can efficiently locate a small set of pixels, without using any gradient information, which when perturbed lead to misclassification by a deep neural network. Our extensive experimental results, somewhat surprisingly, demonstrates the effectiveness of our simple approaches in generating adversarial examples.

Defenses against these attacks is an interesting research direction. However, we note that here that by limiting the perturbation to some pixels (being localized) the adversarial images generated by our local-search based approach do not represent the distribution of the original data. This means for these adversarial images, the use of adversarial training (or fine-tuning), a technique of training (or fine-tuning) networks on adversarial images to build more robust classifiers, is not very effective. In fact, even with adversarial training we noticed that the networks ability to resist new local-search based adversarial attack improves only marginally (on average between 1-2%). On the other hand, we suspect that one possible counter-measure to these localized adversarial attacks could be based on performing a careful analysis of the oracle queries to thwart the attempts to generate an adversarial image.

Finally, we believe that our local-search approach can also be used for attacks against other machine learning systems and can serve as an useful tool in measuring the robustness of these systems.

| Dataset | ErrTop-1 | ErrTop-1(Adv) | CONF | PTB | #PTBPIXELS (%) | TIME (in sec) | Technique | Network |
|---|---|---|---|---|---|---|---|---|
| **NNs trained with batch normalization** | | | | | | | | |
| CIFAR10 | 11.65 | 97.63 | 0.47 | 0.04 | 3.75 | 0.68 | LocSearchAdv (Ours) | NinN |
| CIFAR10 | | 70.69 | 0.55 | 0.20 | 100.00 | 0.01 | FGSM (Goodfellow et al., 2015) | NinN |
| CIFAR10 | 11.62 | 97.51 | 0.74 | 0.04 | 3.16 | 0.78 | LocSearchAdv (Ours) | VGG |
| CIFAR10 | | 11.62 | – | – | – | – | FGSM (Goodfellow et al., 2015) | VGG |
| STL10 | 29.81 | 58.17 | 0.42 | 0.02 | 1.20 | 7.15 | LocSearchAdv (Ours) | NinN |
| STL10 | | 54.85 | 0.53 | 0.20 | 100.00 | 0.03 | FGSM (Goodfellow et al., 2015) | NinN |
| STL10 | 26.50 | 65.76 | 0.47 | 0.02 | 1.11 | 13.90 | LocSearchAdv (Ours) | VGG |
| STL10 | | 26.50 | – | – | – | – | FGSM (Goodfellow et al., 2015) | VGG |
| SVHN | 9.71 | 97.06 | 0.47 | 0.05 | 4.51 | 1.02 | LocSearchAdv (Ours) | NinN |
| SVHN | | 48.62 | 0.49 | 0.20 | 100.00 | 0.02 | FGSM (Goodfellow et al., 2015) | NinN |
| SVHN | 4.77 | 81.10 | 0.66 | 0.07 | 5.43 | 2.15 | LocSearchAdv (Ours) | VGG |
| SVHN | | 4.77 | – | – | – | – | FGSM (Goodfellow et al., 2015) | VGG |
| MNIST | 0.33 | 91.42 | 0.54 | 0.20 | 2.24 | 0.64 | LocSearchAdv (Ours) | NinN |
| MNIST | | 1.65 | 0.58 | 0.20 | 100.00 | 0.02 | FGSM (Goodfellow et al., 2015) | NinN |
| MNIST | 0.44 | 93.48 | 0.63 | 0.21 | 2.20 | 0.64 | LocSearchAdv (Ours) | VGG |
| MNIST | | 0.44 | – | – | – | – | FGSM (Goodfellow et al., 2015) | VGG |
| **NNs trained without batch normalization** | | | | | | | | |
| CIFAR10 | 16.54 | 97.89 | 0.72 | 0.04 | 3.24 | 0.58 | LocSearchAdv (Ours) | NinN |
| CIFAR10 | | 93.67 | 0.93 | 0.20 | 100.00 | 0.02 | FGSM (Goodfellow et al., 2015) | NinN |
| CIFAR10 | 19.79 | 97.98 | 0.77 | 0.04 | 2.99 | 0.72 | LocSearchAdv (Ours) | VGG |
| CIFAR10 | | 90.93 | 0.90 | 0.20 | 100.00 | 0.04 | FGSM (Goodfellow et al., 2015) | VGG |
| STL10 | 35.47 | 52.65 | 0.56 | 0.02 | 1.17 | 6.42 | LocSearchAdv (Ours) | NinN |
| STL10 | | 87.16 | 0.94 | 0.20 | 100.00 | 0.04 | FGSM (Goodfellow et al., 2015) | NinN |
| STL10 | 43.91 | 59.38 | 0.52 | 0.01 | 1.09 | 19.65 | LocSearchAdv (Ours) | VGG |
| STL10 | | 91.36 | 0.93 | 0.20 | 100.00 | 0.10 | FGSM (Goodfellow et al., 2015) | VGG |
| SVHN | 6.15 | 92.31 | 0.68 | 0.05 | 4.34 | 1.06 | LocSearchAdv (Ours) | NinN |
| SVHN | | 73.97 | 0.84 | 0.20 | 100.00 | 0.01 | FGSM (Goodfellow et al., 2015) | NinN |
| SVHN | 7.31 | 88.34 | 0.68 | 0.05 | 4.09 | 1.00 | LocSearchAdv (Ours) | NinN |
| SVHN | | 76.78 | 0.89 | 0.20 | 100.00 | 0.04 | FGSM (Goodfellow et al., 2015) | VGG |

Table 4: Results for four datasets: CIFAR10, STL10, SVHN, and MNIST. The entries denote by denoted by "– " are the cases where the fast-gradient sign method fails to produce any adversarial image in our experimental setup.

| Dataset | ErrTop-1 | ErrTop-1(Adv) | CONF | PTB | #PTBPIXELS (%) | TIME (in sec) | Technique | Network |
|---|---|---|---|---|---|---|---|---|
| ImageNet1000 | 58.27 | 93.59 | 0.29 | 0.29 | 0.43 | 12.72 | LocSearchAdv (Ours) | VGG CNN-S (Caffe) |
| ImageNet1000 | | 85.51 | 0.49 | 1.00 | 100.00 | 4.74 | FGSM (Goodfellow et al., 2015) | VGG CNN-S (Caffe) |
| ImageNet1000 | 58.96 | 91.36 | 0.28 | 0.29 | 0.40 | 10.01 | LocSearchAdv (Ours) | VGG CNN-M (Caffe) |
| ImageNet1000 | | 87.85 | 0.48 | 1.00 | 100.00 | 4.36 | FGSM (Goodfellow et al., 2015) | VGG CNN-M (Caffe) |
| ImageNet1000 | 58.80 | 92.82 | 0.29 | 0.30 | 0.41 | 11.09 | LocSearchAdv (Ours) | VGG CNN-M 2048 (Caffe) |
| ImageNet1000 | | 88.43 | 0.52 | 1.00 | 100.00 | 4.42 | FGSM (Goodfellow et al., 2015) | VGG CNN-M 2048 (Caffe) |
| ImageNet1000 | 46.40 | 72.07 | 0.30 | 0.54 | 0.55 | 73.64 | LocSearchAdv (Ours) | VGG ILSVRC 19 (Caffe) |
| ImageNet1000 | | 85.05 | 0.52 | 1.00 | 100.00 | 23.94 | FGSM (Goodfellow et al., 2015) | VGG ILSVRC 19 (Caffe) |

Table 5: Results for the ImageNet1000 dataset using a center crop of size $224 \times 224$ for each image.

| Dataset | $k$ | ErrTop-$k$ | ErrTop-$k$(Adv) | CONF | PTB | #PTBPIXELS (%) | TIME (in sec) | Network |
|---|---|---|---|---|---|---|---|---|
| CIFAR10 | 1 | 16.54 | 97.89 | 0.72 | 0.04 | 3.24 | 0.58 | NinN |
| CIFAR10 | 2 | 6.88 | 76.65 | 0.88 | 0.07 | 5.50 | 1.02 | NinN |
| CIFAR10 | 3 | 3.58 | 59.02 | 0.90 | 0.08 | 7.09 | 1.85 | NinN |
| CIFAR10 | 4 | 1.84 | 48.89 | 0.90 | 0.09 | 7.63 | 2.12 | NinN |

Table 6: Effect of increasing $k$ on the performance of Algorithm LocSearchAdv (without batch normalization).

ACKNOWLEDGMENTS

The authors would like to thank Hamid Maei for helpful initial discussions.

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
