# Peer review of "Simple Black-Box Adversarial Perturbations for Deep Networks"

_ICLR 2017 — rejected_

[Official Review · AnonReviewer2 · rating 4 · confidence 4 · 14 Dec 2016]
**Blackbox adversarial examples**

The authors propose a method to generate adversarial examples w/o relying on knowledge of the network architecture or network gradients.

The idea has some merit, however, as mentioned by one of the reviewers, the field has been studied widely, including black box setups.

My main concern is that the first set of experiments allows images that are not in image space. The authors acknowledge this fact on page 7 in the first paragraph. In my opinion, this renders these experiments completely meaningless. At the very least, the outcome is not surprising to me at all.

The greedy search procedure remedies this issue. The description of the proposed method is somewhat convoluted. AFAICT, first a candidate set of pixels is generated by using PERT. Then the pixels are perturbed using CYCLIC.
It is not clear why this approach results in good/minimal perturbations as the candidate pixels are found using a large "p" that can result in images outside the image space. The choice of this method does not seem to be motivated by the authors.

In conclusion, while the authors to an interesting investigation and propose a method to generate adversarial images from a black-box network, the overall approach and conclusions seem relatively straight forward. The paper is verbosely written and I feel like the findings could be summarized much more succinctly.

[Official Review · AnonReviewer3 · rating 4 · confidence 4 · 18 Dec 2016]
**review: incremental**

The paper presents a method for generating adversarial input images for a convolutional neural network given only black box access (ability to obtain outputs for chosen inputs, but no access to the network parameters).  However, the notion of adversarial example is somewhat weakened in this setting: it is k-misclassification (ensuring the true label is not a top-k output), instead of misclassification to any desired target label.

A similar black-box setting is examined in Papernot et al. (2016c).  There, black-box access is used to train a substitute for the network, which is then attacked.  Here, black-box access in instead exploited via local search.  The input is perturbed, the resulting change in output scores is examined, and perturbations that push the scores towards k-misclassification are kept.

A major concern with regard to novelty is that this greedy local search procedure is analogous to gradient descent; a numeric approximation (observe change in output for corresponding change in input) is used instead of backpropagation, since one does not have access to the network parameters.  As such, the greedy local search algorithm itself, to which the paper devotes a large amount of discussion, is not surprising and the paper is fairly incremental in terms of technical novelty.

[Official Review · AnonReviewer1 · rating 4 · confidence 3 · 28 Dec 2016]
**Too verbose for little insight**

Paper summary:
This work proposes a new algorithm to generate k-adversarial images by modifying a small fraction of the image pixels and without requiring access to the classification network weight.


Review summary:
The topic of adversarial images generation is of both practical and theoretical interest. This work proposes a new approach to the problem, however the paper suffers from multiple issues. It is too verbose (spending long time on experiments of limited interest); disorganized (detailed description of the main algorithm in sections 4 and 5, yet a key piece is added in the experimental section 6); and more importantly the resulting experiments are of limited interest to the reader, and the main conclusions are left unclear.
This looks like an interesting line of work that has yet to materialize in a good document, it would need significant re-writing to be in good shape for ICLR.


Pros:
* Interesting topic
* Black-box setup is most relevant
* Multiple experiments
* Shows that with flipping only 1~5% of pixels, adversarial images can be created


Cons:
* Too long, yet key details are not well addressed
* Some of the experiments are of little interest
* Main experiments lack key measures or additional baselines
* Limited technical novelty




Quality: the method description and experimental setup leave to be desired. 


Clarity: the text is verbose, somewhat formal, and mostly clear; but could be improved by being more concise.


Originality: I am not aware of another work doing this exact same type of experiments. However the approach and results are not very surprising.


Significance: the work is incremental, the issues in the experiments limit potential impact of this paper.


Specific comments:
* I would suggest to start by making the paper 30%~40% shorter. Reducing the text length, will force to make the argumentation and descriptions more direct, and select only the important experiments.
* Section 4 seems flawed. If the modified single pixel can have values far outside of the [LB, UB] range; then this test sample is clearly outside of the training distribution; and thus it is not surprising that the classifier misbehaves (this would be true for most classifiers, e.g. decision forests or non-linear SVMs). These results would be interesting only if the modified pixel is clamped to the range [LB, UB].
* [LB, UB] is never specified, is it ? How does p = 100, compares to [LB, UB] ? To be of any use, p should be reported in proportion to [LB, UB]
* The modification is done after normalization, is this realistic ? 
* Alg 2, why not clamping to [LB, UB] ?
* Section 6, “implementing algorithm LocSearchAdv”, the text is unclear on how p is adjusted; new variables are added. This is confusion.
* Section 6, what happens if p is _not_ adjusted ? What happens if a simple greedy random search is used (e.g. try 100 times a set of 5 random pixels with value 255) ?
* Section 6, PTB is computed over all pixels ? including the ones not modified ? why is that ? Thus LocSearchAdv PTB value is not directly comparable to FGSM, since it intermingles with #PTBPixels (e.g. “in many cases far less average perturbation” claim).
* Section 6, there is no discussion on the average number of model evaluations. This would be equivalent to the number of requests made to a system that one would try to fool. This number is important to claim the “effectiveness” of such black box attacks. Right now the text only mentions the upper bound of 750 network evaluations. 
* How does the number of network evaluations changes when adjusting or not adjusting p during the optimization ?
* Top-k is claimed as a main point of the paper, yet only one experiment is provided. Please develop more, or tune-down the claims.
* Why is FGSM not effective for batch normalized networks ? Has this been reported before ? Are there other already published techniques that are effective for this scenario ? Comparing to more methods would be interesting.
* If there is little to note from section 4 results, what should be concluded from section 6 ? That is possible to obtain good results by modifying only few pixels ? What about selecting the “top N” largest modified pixels from FGSM ? Would these be enough ? Please develop more the baselines, and the specific conclusions of interest.


Minor comments:
* The is an abuse of footnotes, most of them should be inserted in the main text.
* I would suggest to repeat twice or thrice the meaning of the main variables used (e.g. p, r, LB, UB)
* Table 1,2,3 should be figures
* Last line of first paragraph of section 6 is uninformative.
* Very tiny -> small

[Final Decision · Program Chairs · 06 Feb 2017]
**ICLR committee final decision**

While this is an interesting topic, both the method description and experimental setup could be improved.